# Data Pricing Mechanism Based on Property Rights Compensation Distribution Conference Submissions

## Abstract

While machine learning (ML) benefits from data, it also faces the challenges of ambiguous data ownership, including privacy violations and increased costs of using data. This suggests that the value created by data is determined not only by its utility but also by the cost of using the data (negative externalities). The existing pricing methods mainly value data based on its utility but ignore the negative externalities caused by fuzzy ownership, therefore can not design an efficient pricing mechanism. Throughout the data life cycle (creation, pre-processing, training, etc.), the usufruct and ownership of the data are transferred at the same time, so the benefits and costs are generated simultaneously. Considering that data rights confirmation and data pricing cannot be separated independently in the process of data transaction, we propose the first data valuation mechanism based on modern property rights theory in this paper. Specifically, we propose to clarify the ownership of property rights through the integration of property rights and improve the final revenue of the whole workflow "data link" through the form of the whole collective, while compensating process performers who lose ownership after the integration. Then, we consider the expectations of both the integrator and the integrated party during the compensation allocation. For the former, we apply compound interest to assess a total compensation equivalent to the time value for the Data chain. For the latter, we respect and meet their expectations as much as possible. To achieve this, we provide the framework based on Least-core to assign the compensation and prove that our framework can also work compared to existing algorithms. Finally, to cope with more complex situations, we adjust the traditional Least-core and demonstrate theoretically and experimentally that the compensation mechanism is feasible and effective in solving the data pricing problem.

## 1 Introductioin

While ML benefits from data, it also faces challenges brought by the ambiguity of data ownership (Maini et al., 2021). According to the life cycle, data is generated by the producer and then passes through agents such as data preprocessors and model pre-trainers, and finally generates value through model training. In this process, data ownership and access are transferred simultaneously, which makes the utilization of data have a cost: On the one hand, this makes the subject of privacy protection unclear, posing the potential for privacy violations. The Cambridge Analytica scandal, for example, where Facebook and Cambridge Analytica collected the personal data of up to 87 million Facebook users without their consent (Kelly), gave rise to an unprecedented discussion on data privacy. On the other hand, even though a large number of data pricing models are already available in the data marketplace to value data (Chen et al., 2019; Liu, 2020; Li et al., 2013; Koutris et al., 2013; 2015), each use of a dataset requires a valuation, since datasets behave differently in different models. This increases the negative externalities of ML (e.g., huge computational volumes).

The imbalance between the benefits of data and the costs of using it makes clarifying property rights an important issue. Demsetz (1967) points out that the generation of property rights is essentially a process of cost-benefit tradeoff. Property rights arise when the benefits of internalizing externalities by defining property rights are greater than the costs of engaging in the act. Furthermore, the purpose

of clarifying property rights is to maximize returns by internalizing externalities. Taking the data life cycle as an example, the concrete task of confirming the ownership is to delineate the controllable rights and clarify the owner of the rights (Asswad & Marx Gómez, 2021), whose purpose is to internalize externalities to the maximum extent. In other problem setting, such as feature selection or data source valuation, giving the most valuable feature or data source ownership can internalize externalities ( e.g., reduce computations after setting the appropriate utility function ) and improve the performance. It should be noted that the revenue of data transactions under the premise of data ownership includes both traditional revenue and the cost caused by negative externalities that need to be subtracted. And the purpose of rights confirmation is to maximize the revenue of data at this time.

The existing data marketplace always assumes that the ownership of data belongs to its producer, and attributes the costs of data transactions to privacy protection, making data privacy protection becomes an issue juxtaposed with data pricing rather than unified. Specifically, traditional data pricing schemes including data-based pricing, model-based pricing (Chen et al., 2019; Liu, 2020), and query-based pricing (Li et al., 2013; Koutris et al., 2013; 2015), have paid varied degrees of attention to avoiding privacy leaks. The marketplace with data-based pricing allows the customers to access datasets entries directly, which makes it challenging for ownership protection. To address this challenge, the model-based pricing framework propose creating different query versions by carefully adding different noise (Chen et al., 2019) or sells a series of differentially private models to respect data owners' privacy restrictions (Liu, 2020). In terms of the query-based pricing framework, it makes decisions about the restrictions on data usage (Li et al., 2013), which partially alleviate the shortcomings of privacy protection. However, their pricing mechanism still reflects the value of the data by quantifying the utility of the data or quantifying the model training examples, which, incidentally, avoids the data owner's privacy from being breached. Such passive protection can only strive to prevent privacy leaks but fail to integrate more negative externalities.

In this case, we propose a data pricing method based on property rights compensation, which is different from the previous pricing mechanism. Our contributions are mainly reflected in:

- Clarifying the data ownership. Through the introduction of modern property rights theory, we answer the question of maximizing the internalization of externality: integrate the cooperative parties into a whole, and the one who makes the most marginal contribution holds the overall data ownership. In fact, according to the logic of modern property rights theory, ownership can be determined for all agents with cooperative relations through integration. The effect of it will be improved with the increase of marginal contribution difference of agents, which can extend our framework to the whole life cycle of data. In this paper, we take the background of feature selection and data source pricing as an example to prove the feasibility of the method.

- Discussing the Data pricing from the perspective of compensation that the transfer of data property rights necessitates the payment of compensation. Since the use of data brings both benefits and costs, we propose that the valuation of data should not only be based on benefits but also consider its costs. Although the cost cannot be directly quantified, we internalized externalities furthest through property rights integration to maximize the value of the data. On this basis, the compensation for distorted ownership is estimated by the time value of the data. Experimental results show that this method can still complete the task of data pricing.

- We propose using the least-core rather than other concepts in cooperative games to solve the allocation scheme. Since the process of ownership integration requires the integrated party to transfer ownership to form a grand coalition, the withdrawal of either agent will increase the negative externalities of the coalition. We are more inclined to realize the stability of the distribution scheme through the core. In this process, we balance the expected compensation of both the integrator and the integrated party. Finally, we discuss the feasibility of this framework under different conditions by adjusting the coefficients of the deficit parameter.

## 2 RELATED WORK

**Data ownership in Machine Learning.** Research in the field of machine learning attaches some importance to the ownership of data or models. Specifically, the work revolves around ownership utilization and protection. For the former, the work of Maini et al. (2021) has achieved to determine whether a potentially stolen model was derived from an owner's model or dataset. And for the latter, the existing measures for protecting data ownership in the data marketplace include legal means and technical means.

For the former, countries have been experimenting with relying on the traditional model of "empowerment-violation" to provide a legal basis for data property rights protection, and have achieved some practical results, such as the well-known General Data Protection Regulation(GDPR) and the California Consumer Protection Act(CCPA). Using the GDPR as one of the patterns, the producers of data are endowed with data ownership. The existing law regulates data rights and legal obligations. Although, the law provides a powerful weapon for data property rights, it continue to face constraints in practical application. For instance, GDPR highlighting that it may fundamentally change the way big data analysis is done, making it a suboptimal and inefficient method of protection (Zarsky, 2017). Additionally, the number of cases filed by parties is limited and is difficult to obtain satisfaction of a claim in real life. Thus, the protection of data ownership has not achieved the expected performance despite rapid legislative progress (Li et al., 2019). The legislation has established a legal basis for the ownership issue, but it is still unable to protect it favorably. This poses a significant hidden risk to the establishment of a healthy data trading market.

For the latter, the fundamental technical solution is to shift from centralized paradigm to decentralized paradigm. And the key technologies to the decentralized paradigm include Federated learning, Secure Multi-Party Computation (MPC), Blockchain, etc. In federated learning, clients collaboratively train a shared model under the orchestration of central server, while keeping the training data decentralized (Li et al., 2020). Due to its advantages in privacy protection, there are increasing works that have been proposed to use federated learning in healthcare (Xu & Wang, 2021), city management (Jiang et al., 2020), etc. MPC enables participants jointly compute a function with their data while keeping data private. A tamper-proof ledger using the blockchain may be used to record digital interactions (Wirth & Kolain, 2018). These interactions not only contain operations on the data but can also record the process of data modeling and analysis (Wang et al., 2018). If the complexity of these technologies is not taken into account, we could embed privacy into the entire engineering process, with these technologies effectively involved in the loop (Martín & Kung, 2018).

**Compensation distribution.** There are a number of compensation distribution criteria proposed from the area of cooperative games. The acquiescent baseline to assign the data importance/value to a model is leave-one-out (LOO). Another conception, Shapley value (Shapley, 1953), is the most common one amongst all of the valuation criteria. It satisfies some proven theoretical properties, provides an efficient and fair solution to distribute contributions among players (clients or participants) by assigning to each real value which denotes its influence (profit). It is Ghorbani & Zou (2019) who develops a principle framework to address data valuation in the context of supervised machine learning. More precisely, given a learning algorithm trained on $n$ data points to produce a predictor, data Shapley was first proposed to quantify the value of each training datum to the predictor performance. Besides, to solve this question of fairly distributing profits among multiple data contributors, Jia et al. (2019) proposed a novel approach, which addresses the problem of data valuation by utilizing the Shapley value. Furthermore, Kwon et al. (2021) derives the first analytic expressions for DShapley for the canonical problems, and proposed an algorithm several orders of magnitude faster than D-shapley.

## 3 PRELIMINARIES AND BACKGROUND

In this section, we introduce the background and preliminaries of the framework. First, we introduce the definition and generation of the data ownership to help understand the importance of clarifying property rights. Then we formalize the data workflow and introduce the theory to identify ownership. Finally, we introduce the solution to distributing the compensation. The notations used frequently are showed in Table 1.

Table 1: The summary of notations

| NOTATION | DEFINITION |
| --- | --- |
| $\check{D}C$ | Data chain |
| $DP$ | data product (performance of the machine learning) |
| $A_i$ | the $i^{th}$ action of Data chain |
| $a_i$ | the agent conducting the $i^{th}$ action |
| $W_i$ | the expected compensation of the $i^{th}$ agent |
| $C_t^i$ | the time value of the $i^{th}$ agent at time $t$ |
| $B_t$ | the compensation for $DC$ estimated by the integrator at time $t$ |
| $B_t^i$ | the compensation for the $i^{th}$ agent estimated by the integrator at time $t$ |
| $b_t^i$ | the payoff of the $i^{th}$ agent at time $t$ |
| $\phi_i$ | the compensation allocated to the $i^{th}$ agent |
| $\psi_i$ | the contribution of the $i^{th}$ agent |
| $\gamma_i$ | the contribution rate of the $i^{th}$ agent |
| $d$ | the deficit parameter in least core |

## 3.1 OWNERSHIP OF DATA

Despite the absence of a unified view on the definition of the data ownership, it is widely agree that the rights of it including, but not limited to access, creation, generation, modification, analysis, use, sell, or deletion of the data, in addition to the right to grant rights over the data to others (Asswad & Gómez, 2021). If we explain the generation of data property rights according to the theory of property rights, it shows that the benefits of internalizing externalities in the process of data value extraction have been greater than the costs of direct stepwise training data. The property right of an asset needs to be defined either because the income of defining the ownership becomes larger, or because the cost of it becomes smaller. The reasons for the emergence of data property rights include the above two aspects: on the one hand, with the widespread usage of new information and communications technologies (ICTs) such as highspeed networks, cloud computing, and pervasive data collection (Hashem et al., 2015), we now have a vast amount of data available. At the same time, upgrades in computer hardware and improved algorithms have substantially increased the power of analytical techniques, and thus the value of data has been better derived. On the other hand, the distributed frameworks including blockchain also provide a way to protect owners from the transfer, reducing the cost of defining ownership.

## 3.2 DATA CHAIN

Data has value, but the value we can get from data is non-linear and unpredictable as: i) The utility of a data-driven product or service depends not only on the data, but also on the model upon data, making it difficult to understand the value of data before modeling. ii) The value of data can be augmented with other data, in a nonlinear manner. iii) The utility of a data-driven product of service also depends on the user of it. To this end, we propose that the realization of data value extraction needs a workflow. It is in this workflow that the process of data circulation is accompanied by the movement of data ownership and usufruct of it.

**Definition 1.** A data chain $DC$ is a consequence of actions $(A_1, A_2, ..., A_n)$ upon data to create a data product $DP$, which creates payoff $B_t$ at time $t$.

## 3.3 THEORY OF MODERN PROPERTY RIGHT

We introduces Modern Property Rights Theory in the field of economics to provide a theoretical basis for ownership integration in the Data chain. After integration, the property rights are further clarified. In this subsection, we introduce the background and basic logic of modern property rights theory, and in the subsection4.2, we explain how to integrate the ownership of $DC$.

**The modern propety rights theory.** This theory identifies ownership with residual control rights and argues that, in a setting with contracts that are incomplete and cannot mandate investment decisions, firm boundaries are determined by which ex ante investments are most valuable. It is worth noting that the concept of an incomplete contract was first introduced in the seminal work of modern property rights theory, which elaborated that a contract cannot be transformed into a comprehensive and complete contract through meticulous enumeration. The author, Grossman & Hart (1986) stated that the essence of ownership is the right to control the subject matter, with all exceptions except the contract's content. Since then, a number of articles have expanded on this concept, gradually forming the modern theory of property rights (Hart & Moore, 1990; 1991; Hart et al., 1996). The basic idea is that property rights created through collaboration should be assigned to the party who contributes the most to the post-cooperation output. In comparison to the transaction cost theory (Coase, 1960), this theory introduces a new concept for vertical integration and addresses the question of "who integrates" during the integration process.

**The necessity of integration.** The modern property rights theory extends the definition of the property rights, which can be of two types: specific rights and residual rights. Data ownership refer to the control over the remaining property rights in addition to those specified in the contract. However, in the data marketplace, simply entrusting data property rights to them (such as individuals and small companies, etc.) is uneconomical. Externalities will greatly increase the operation cost of the data market and hinder its development. The externalities during different stages in a $DC$ include but are not limited to:

- Difficulties in data collection. The data collected from each original owner involves negotiation.

- The high cost of model training. Both the time and space costs of data processing and the evaluation of data value increase exponentially with the volume of data. As we proposed the strategy network pre-training earlier in order to alleviate the time complexity with $O(2^n)$ .

At this point, the advantages of ownership integration are obvious: On the one hand, the externalities stated above are internalized through vertical integration of property rights in $DC$, which will greatly reduce the costs and energize the data market. On the other hand, allowing a particular agent to integrate property rights makes production decisions more uniform. At the same time, sub-optimal Nash equilibria resulting from each player's exclusive investment will no longer occur.

### 3.4 COMPENSATION ALLOCATION CRITERIA

After integrating the property rights of the $DC$, we hope to establish a stable allocation scheme, which can simultaneously satisfy the expectation of all agents. There are a number of allocation criteria proposed from the cooperative game. Shapley value, the most famous criterion we have introduced above, is more concerned with the fairness of the allocation instead of stability. Intuitively, we should adopt a criterion with more concern about the stability of the distribution to make the consolidated $DC$ more stable. What worse, it has been demonstrated that calculating approximate Shapley value in the context of big data fails to obtain approximate from samples with a uniform distribution (Balkanski et al., 2017). Therefore, we propose to use Core instead, which has not only an intuitive superiority over shapley value for solving compensation allocation in the context of property rights consolidation, but also a theoretical superiority in terms of the approximation algorithm's convergence guarantee. Furthermore, another cooperative game solution set that also involves the perspective of both parties should be concerned. The bargaining set (Aumann & Maschler, 2016) is defined similarly to the core, while Einy et al. (1999) demonstrated that both core and least-core are included in the bargaining solution set. However, the bargaining solution set only considers coalition deviations that are stable in themselves compared to the core, i.e. no inverse deviation is permitted, which contradicts the complexity expected from actual compensation.

**Core and the least-core.** Consider the characteristic function game $G = (N, v)$ and its outcome $(CS, x)$. If the total payoff of a coalition $S$ under $x$ is denoted by $x(S)$, and $x(S) \leq v(S)$ for some $S \subseteq N$, the agents in $S$ might benefit more from departing the coalition structure $CS$ and forming their own coalition. This gives each member of $S$ an incentive to deviate. As a result, the outcome $(CS, x)$ is unstable. The core of $G$ is the set of stable outcomes, distributing the outcomes where no subset of agents has an incentive to deviate. Therefore, the core could be defined as follow:

**Definition 2.** The core of a characteristic function game $G = (N, v)$ is the set of all outcomes $(CS, x)$ such that $x(S) \geq v(S)$ for every $S \subseteq N$.

However, the stable coalition structure may not exist. To assure a non-empty solution set, we naturally request that the core notion be relaxed, requiring merely that no coalition gain considerably from deviations(if not, using Shapley value to distribute compensation will also make Data chain $(DC)$ unstable), i.e. no member deviations. We assume that each agent has an expected $W_i$ for their compensation, which is the bottom line when the $a^*$ allocates compensation. In this vein, adding an appropriate relaxation parameter $d$ can alleviate the problem. This gives rise to the following definition:

$$\sum_{i \in S} \phi_i + d \geq \sum_{i \in S} W_i \quad \forall S \subseteq DC \tag{1}$$

Then, any cooperative game would not has an empty core as long as the $d$ is large enough. When $d$ is positive, it means that there is a gap of expected compensation between the integrated and the integrating agent. Then we hope to minimize this gap so that the property rights of $DC$ can be integrated in the most stable way. Inversely, the negative $d$ means that the integrating agent has paid compensation that exceeds the expected amount of the integrated. In this case, minimizing the additional compensation could reduce the costs while meeting the needs of the integrated agents. Therefore, we are interested in finding the smallest $d$, which would be defined as a set of solutions to the following linear program:

$$\min \quad d \tag{2}$$

$$s.t. \quad \begin{aligned} &\sum_{i \in N} \phi_i = B \\ &\sum_{i \in S} \phi_i + d \geq \sum_{i \in S} W_i \quad \forall S \subseteq DC \end{aligned} \tag{3}$$

The the time complexity of exact solution of the least core is $O(2^N)$. Even calculating this seemingly simple linear programming equation faces the same amount of computation as shapley value. But it turns out to be possible to find the least core solution with probability at least $1 - \delta$ (Yan & Procaccia, 2021).

$$\Pr_{S \sim CD} [\phi_i + e^* \geq W_i] \geq 1 - \delta \tag{4}$$

Since the Balkanski et al. (2017) points out that the generalization error of the PAC-learnable function is small when the number of samples is sufficient, and the solution under arbitrary precision can be calculated from the number of samples related to the VC-dimension, we can give the $\delta - probable\ least\ core$ a probabilistic guarantee.

**Theorem 1.** Given $\delta, \Delta > 0$, the linear program over $O((n + log(1/\Delta))/\delta^2)$ coalitions sampled from $DC$ solved by the Monte Carlo algorithm would give a payoff allocation in the $\delta - probable\ least\ core$ with probability at least $1 - \Delta$.

The proof of theorem 1 will be provided in the appendix A.

**Compound interest** How can the property rights consolidation process ensure that the property rights are not undervalued? The propose of this subsection is to estimate the payoff to each action (and its providers).

The compound interest considers the time value of the money. And the time value of money reflects that currently owned currency has greater value than the same amount of currency received in the future, thus the currency can be invested and compounded. Since the data is a special type of asset (Bodendorf et al., 2022), we naturally apply the compound interest to estimate the time value of the data. According to this concept, the interest will be calculated based on the principal amount in addition to the newly obtained interest can also generate interest. In the context of data trading market applications, the core theoretical basis of an agent's capital accumulation is the principle of compound interest, which allows one to obtain a desirable and significant time value by investing over a long period. Taking the initial moment of investment $b_0^i$ of $agent_i$ as an example, the time value after $t$ times investments is calculated as follows:

$$C_t^i = b_0^i \cdot \gamma_t^i, \tag{5}$$

with compound interest $\gamma_t^i = \left(1 + \frac{\gamma_0^i}{t}\right)^t$. As a result, when $a_i$ completes the first phase of its investment, the principal amount is expected to increase to $b_0^i \cdot (1 + \gamma_0^i)$. It is worth noting that

after completing the second period of investment, $agent_i$ 's principal is expected to increase to $b_0^i \cdot (1 + \frac{\gamma_0^i}{2})^2$ rather than $b_0^i \cdot (1 + \gamma_0^i)^2$ due to the subsequent discount in the nominal interest rate as the investment period increases. When the number of investments approaches infinity, the time value is calculated in a continuous form as follows:

$$\lim_{t \to \infty} (1 + \frac{\gamma_0^i}{t})^t = e^{\gamma_0^i}. \tag{6}$$

## 4 METHOD

In this section, we will explain how to finish the data pricing task by compensation distribution. The framework proposes to integrate the ownership according to the contribution rate first, dividing the agents of the Data chain into integrator and integrated party. After that, it takes both the two parties' expectations into account to allocate the compensation.

### 4.1 AGENT'S CONTRIBUTION FOR DATA CHAIN

To evaluate the actions in a data chain, we assume the payoff $B_t$ for the data chain at the time $t$ is given. Then the problem is how to quantify the contribution to all the participating agents in $DC$. We then propose a revised version of Shapley Value. Specifically, we define a baseline provider $a_i^*$ for $A_i$, which is the provider with no extra cost to serve the baseline function of $a_i$. For example, for data providers, the baseline provider can be the open data with no cost; and for data modeling, the baseline provider can be the open-sourced pre-trained model without extra cost for model training. Therefore, the coalition would prefer a low-cost baseline compared to the agent with negative contribution rate. Specifically, the contribution $\psi_i$ and the contribution rate $\gamma_i$ of $a_i$ are defined as:

$$\psi_i = \sum_{S \subseteq DC/i} \frac{|S|!(|DC| - |S| - 1)!}{|DC|!} (v(S \cup a_i) - v(S)), \quad \gamma_i = \frac{e^{\psi_i}}{\sum_i e^{\psi_i}} \tag{7}$$

where $v(S \cup a_i) - v(S)$ is actually about the difference of utilities of two data chain, one with $a_i$ and the other without $a_i$ but having a baseline provider $a_i^*$ instead. We prefer softmax to the linear normalization for the following two characters: First, a negative marginal contribution can be normalized to a positive one by the softmax function. This avoids a special case where linear normalization cannot be calculated when the Shapley value of two agents is 0.5, -0.5 respectively. Secondly, when the number of agents is small, the difference in the marginal contribution can be huge. The linear normalization will cause the value of b to be close to 0 or 1, while the softmax normalization can alleviate this phenomenon. When the number of agents is large, the two normalization methods are similar to each other. Our experiments on synthetic data sets illustrate the second feature of the softmax function.

### 4.2 PROPERTY RIGHTS INTEGRATION

According to the modern propety rights theory, the goal of data ownership integration is to maximize the value generated by the $DC$. The ownership or residual control over data beyond the user agreement in the data integration process should be shifted to the agent that plays a key role, in order to incentivize the agent to invest more resources in the data market's efficient operation. To realize the data property rights transaction in the context of modern property rights theory, the agent that contributes more significantly, i.e., the party that integrates user data, only needs to pay "compensation" to the user during the transaction process. Accordingly, we propose that $a^* = a_{\underset{i \in N}{argmax} \gamma_i}$

should integrate the other agents in the $DC$. This idea is inspired by modern property rights theory.

For example, if $a^*$ has the largest marginal contribution, he would have the largest unit input-output for $DC$. At this point, $a^*$ can integrate other $agents$ to obtain their control. It means that $a^*$ obtains the power to make the decision of all proprietary investment through integration, and guarantees output optimization through residual control. Then $a^*$ would gain the "income" of property rights. But correspondingly, other $agents$ also lose investment motivation, resulting in insufficient investment and distorting the final output. However, since the investment of $a^*$ is more important to $DC$,

the distortion will be much smaller than the efficiency loss of Nash equilibrium. The larger the difference of marginal contribution is, the closer the integration result will be to the optimal output.

Specifically, taking the $DC$ with two agents as an example. Suppose that $a_1$ is the company that preprocesses the data and $a_2$ is the company that trains the model. Let $q_i(i = 1, 2)$ to be the sufficiently complex production decisions which cannot be specified completely in an initial contract between the firms. $a_1$ and $a_2$ represent the exclusive investments of $a_1$ and $a_2$, respectively. $DP$ represents the output after integration, and here refers to the performance of modeling. To maximize the output of the $DC$, more accurate models that meeting the requirements of the order should be trained. As is mentioned above, the $DP$ may not be ideal before integration. Meanwhile, $q_1$ (data quality) and $q_2$ (the cost of model training) cannot be reflected in advance (i.e., in the contract). Therefore, the noncontractibility $q$ leads to the need to allocate residual control rights. Then the integration may occur in two situations: i) When $a_1$ owns $a_2$, it can exercise the control over $a_2$ to improve the training accuracy of the model. ii) When $a_2$ integrates $a_1$, it can exercise control over $a_1$ to increase the preprocessing input. If $a_2$ has a greater marginal contribution, $a_1$ will be consolidated and its proprietary investment in $period_1$ will be controlled by $a_2$ to optimize $DP$.

### 4.3 COMPENSATION DISTRIBUTION

**On the integrator perspective.** The agent who contributes most to the $DC$ owns the property rights and therefore gains the income from now on. Since the compensation paid for the integrated party becomes the cost for him, it is naturally expected to be as few as possible. But no matter how low the compensation is, it should be at least equal to the time value of the pre-integration capital at any time in the future, motivating the other agents willing to lose control. As a result, the total compensation paid by the consolidator to all other $agents$ should satisfy the following condition:

$$B_0 = \sum_{i \in N} B_0^i, \tag{8}$$

with $\quad \lim_{t \to \infty} \frac{C_t^i}{B_0^i} = 1.$

**On the integrated perspective.** We define $W_i$ to be the compensation expected by the $a_i$ who loses residual control. In this paper, we argue that the rational expectation should be consistent with the time value that can be calculated by Eqn(5). Inversely, irrational expectation may be either too large or too small in absolute value, or the relative expectation ranking among $agents$ may not match the contribution ranking.

**Allocation with least-core.** Although our least-core scheme can provide a distribution scheme that is most conducive to the alliance's stability, it does not mean that disregard fairness entirely is reasonable. The aim of the distribution is to allocate the compensation based on the time value of $DC$ at time $t$ as far as possible to meet the expectations of each $agent$. We can still allocate according to the contribution rate in order to make the distribution in the irrational case. To achieve this goal, we adjust the deficit parameter $d$ under different coalitions by adding the coefficient $c$, which can be defintie as follow:

$$\min \quad d \tag{9}$$

$$s.t. \quad \begin{matrix} \sum_{i \in N} & \phi_i = B_0, \\ \sum_{i \in S} & \phi_i + c_S d \geq \sum_{i \in S} & W_i, \quad \forall S \subseteq N, \end{matrix} \tag{10}$$

with $\quad c_S = \sum_{i \in S} e^{-\gamma_0^i}.$

It should be noted that: i) when the marginal contribution rate of each agent is the same, the linear programming degenerates to the regular least-core solution2. ii) when $W_i$ is constant in different coalition $S$, the number of inequality constraints can be reduced to $|N|$. The calculation amount of equations (9) and (10) will mainly come from the marginal contribution rate $\gamma^i$. As we use tmc-Shapley (Ghorbani & Zou, 2019) to estimate SV, the calculation amount is at most $O(N^2)$ according to Theorem 1. We shall provide theoretical justification for the reasonableness of this change in the appendix.

## 5 EMPIRICAL EXPERIMENT

Throughout the experiments, we are trying to understand the following: i) Would the proposed Least-core be approximated by sampling according to **Theorem 1**? ii) Could the proposed Least-core gain benefits compared to the classical valuation criteria for valuation problems? iii) To what extent can least-core be competent for allocation under irrational expectations. The following experiments are conducted on feature valuation and data valuation tasks. Although Yan & Procaccia (2021) has demonstrated the feasibility of core as a replacement for shapley value, our work adjusts the deficit parameter $e$ and the inequality constraint functions are random. Since it is infeasible to train a logistic regression classifier on all possible subsets of large-sized datasets as well as get the exact Shapley values and least core, we verify **Theorem 1** in the context of the samll-sized problem(such as feature valuation). For the other two experiments, we verify in the context of data valuation, because large-scale datasets are closer to real-world application.

### 5.1 EXPERIMENT ON FEATURE VALUATIONS

Since the assessment of the data full cycle involves fewer stages (such as data pre-processor,model trainer, etc.). We hope to experiment on larger datasets to demonstrate the feasibility of our proposed method compared with the baseline. Therefore, we choose the feature selection and data valuation of the problem setting. We choose the wine datasets with 13-class classification and 178 samples (`https://archive.ics.uci.edu/ml/machine-learning-databases/wine/wine.data`). And the results of the same experiment on the bigger datasets including MNIST, CIFAR-10/100 would show in the appendix later. Let the agents to be the features and the define expected compensation of a coalition by equation 7. To empirically verify **Theorem 1**, we sample the fraction of coalitions uniformly at random from all possible coalitions, and compute the least core by the Equation 2 to these coalitions. We then exams what fraction of all coalitions satisfy the least core constraints with respect to the exact deficit $e$. These two kinds of fraction help us to get the accuracy $1-\delta$ and thus lead to $\delta-probable\ least\ core$. As can be seen in Figure 2(left), even with restriction less than 10% could achieve prediction accuracy up to 80%.

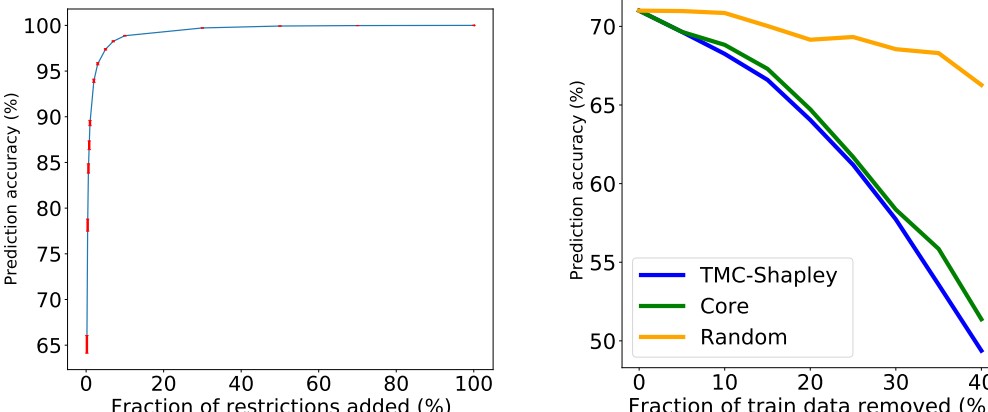

Figure 1: Least core accuracy (satisfaction of the core constraint) over coalitions(left). Data removal results of Synthetic data(right).

### 5.2 EXPERIMENT ON DATA VALUATIONS

First, We emulate the data removal experiments on Synthetic data as described in (Ghorbani & Zou, 2019) to verify that the proposed least-core is up to the task of valuation. Concretely, we conduct data removal: Sorting the data from most valuable to least valuable, eliminating the most valuable valuable 5% of the data at a time. Intuitively, the best criteria would induce the fastest drop of performance. As we can see in Figure 2(right), the performance of least-core is as well as that of TMC-shapley.

Furthermore, we divide the irrational expectation into two cases: i) the expected compensation exceeds or falls below the time value of the agent; ii) the order of the agent's expected compensation in the $DC$ does not match the order of its contribution. To verify the effect of revised Least-core on case i, we use a Gaussian random distribution noise (mean=0, standard deviation $= 0.1, 1$) injection strategy that perturbates the $W_i$ to simulate irrational expectation of compensation. For another case, we randomly selected $10\%, 30\%$ of all the agents to disrupt the order of $W_i$, while other variables remain unchanged. And this time, we add breast cancer datasets from the UCI Machine Learning repository(`https://goo.gl/U2Uwz2`). Figure 3(left) illustrates that the accuracy drops down while the deviation increase. And Figure 3(right) shows that the revised version alleviates the impact of the irrational expectation while does not interfere with the solutions under rational expectations.

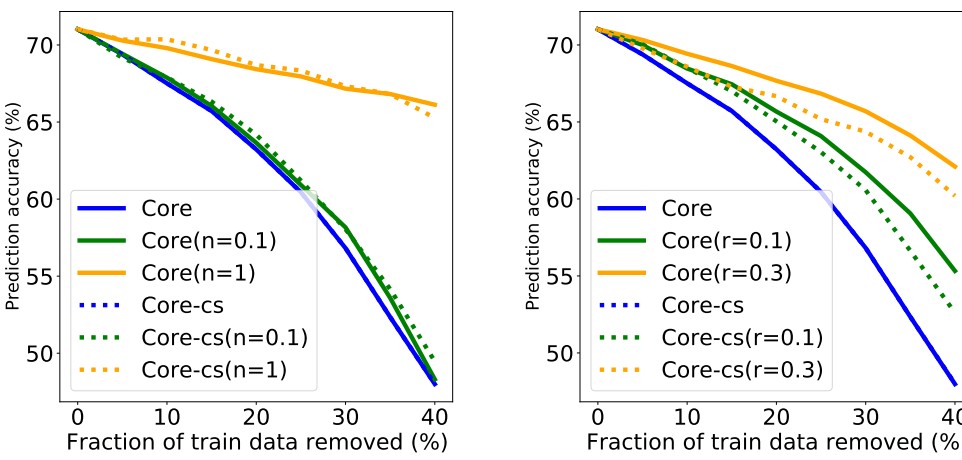

Figure 2: Data remove result: Curves of logistic regression test performance after injecting Gaussian random distribution with standard deviation $= 0.1, 1$(left) and disrupting the order of $0.1, 0.3$ agents (right). The best (Synthetic) data points ranked according to the least-core and the least-core with coefficient $c_S$ are removed.

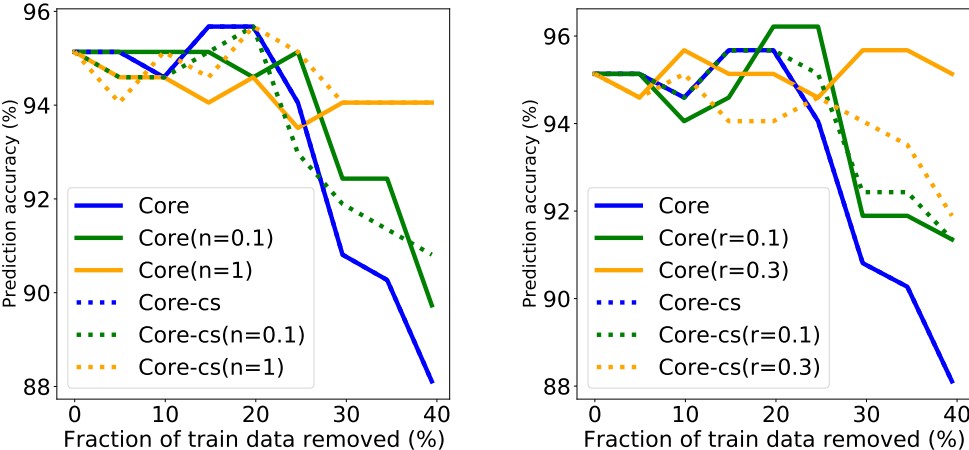

Figure 3: Data remove result: Curves of logistic regression test performance after injecting Gaussian random distribution with standard deviation $= 0.1, 1$(left) and disrupting the order of $0.1, 0.3$ agents (right). The best (Natural) data points ranked according to the least-core and the least-core with coefficient $c_S$ are removed.

## 6 CONCLUSION AND FUTURE WORK

We have presented a data valuation mechanism based on modern property rights theory. The framework responds to the requirements of data rights confirmation in the data marketplace and provides a scheme for quantifying property rights compensation. The theoretical and empirical results suggest the our framework is a principled means of doing credit assignment in ML whether the demand of the integrated party is rational or not.

There are several exciting directions for future work. First, multiple agents can co-exist in practical applications, which form a competitive relationship to maximize the revenue for themselves. Second, a criteria to determine the most suitable adjustment coefficient while dealing with the irrational expectation of the integrated party should be given.

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

## A   THE PROOF OF THEOREM 1

We employ the following known lemmas (Balkanski et al., 2017) to prove that the least-core could be approximated by sampling.

**Lemma 1.** Denote $H : \mathbb{X} \to \{-1, 1\}$ with $n$ VC-dimension. If function $f$ is the groundtruth and the number of samples $m = O(\frac{(n + log(\frac{1}{\Delta}))}{\delta^2})$. Then,

$$|\underset{S \sim DC}{P}[h(x) \neq f(x)] - \frac{1}{m}\sum \mathbf{1}_{h(x^i)} \neq f(x^i)| \leq \delta$$

where $x^1, x^2, ..., x^m \sim DC$.

**Lemma 2.** The function class $\{x :\mapsto sign(w \cdot x)|w \in \mathbb{R}^n\}$ has VC-dimension $n$.

According to the lemma 2, we denote $h \in H$ is the class of function that satisfies: $H = \{h :\mapsto sign((\phi, 1, e) \cdot (\mathbf{1}_{i \in S}, -W_i, 1)), \phi_i \in \mathbb{R}^n, e \in \mathbb{R}, \sum_{i \in N} \phi_i = B\}$. This allows $H$ to be a subset with $n + 2$ VC-dimension containing the least-core. We then define that $\phi_i^*, e^*, f^*$ respectly represent the compensation allocation in the least core, the $e$ subsidy and the classifier accordingly.

Let vector $y^S = (\mathbf{1}_{i \in S}, -W_i, 1)$, then $h(y^S) = 1$, for all $i \in S$, and $f^*(y^S) = 1$ for $S \in 2^N$.

$$\underset{S \sim DC}{P}[\phi_i - v(S) + e^* \geq 0] \geq \underset{S \sim DC}{P}[\phi_i - v(S) + e \geq 0] \tag{11}$$

$$= 1 - \underset{S \sim DC}{P}[\hat{\phi}_i - v(S) + \hat{e} \geq 0] \tag{12}$$

$$= 1 - \underset{S \sim DC}{P}[\hat{h}(y^s) \neq f^*(y^s)] \tag{13}$$

$$= 1 - (\underset{S \sim DC}{P}[\hat{h}(y^s) \neq f^*(y^s)] - \frac{1}{m}\sum \mathbf{1}_{\hat{h}(y^{s^i}) \neq f^*(y^{s^i})}) \tag{14}$$

$$\geq 1 - \delta \tag{15}$$

## B   COMPUTATION OF THE LEAST-CORE

Actually, the equation9 may be an overdetermined linear equations. Thus, the exact distribution $\Phi = (\phi_1, ..., \phi_N)$ is a least square solution from the following equation:

$$\min \quad \|\Phi\|^2 \tag{16}$$

$$s.t. \quad \begin{aligned} &\sum_{i \in N} \phi_i = B_0, \\ &\sum_{i \in S} \phi_i + c_S \cdot e \geq \sum_{i \in S} W_i, \quad \forall S \subseteq N, \end{aligned} \tag{17}$$

with $e$ calculated from equation9.

## C  INTERPRETATION OF ADJUSTMENT PARAMETER

To proof that the solution set of least-core with adjusted parameter $e$ is still stable, we empoly the boundedness principle to prove that perturbation error is controllable as well as the Proposition from Kwon & Zou (2022). For the benefit of modifying $e$, we give a formal proof in an easy-to-understand graphical way.

### C.1  ERROR ANALYSIS

Studying the influence of adding coefficients on linear programming problems is actually a perturbation error problem.

**Lemma 3.** When the cardinality $|S|$ is large enough, the performance change $U(S \bigcup z^*) - U(S)$ is near zero, and thus the marginal contribution $\Delta|S|(z^*; U, D)$ becomes negligible for any $z^*$.

Therefore, the marginal contribution of $Agents$ is bounded for a coalition under any cooperative combination, the same is true as the contribution rate7. We then denote the supremum of marginal contribution rate to be $\gamma_{max}$. To distinguish between the natural index $e$ and the deficit parameter, we define the latter as $\hat{e}$. The coefficient $c_S$ of any inequality constrained deficit parameter $\hat{e}$ satisfies $c_S \leq \hat{e}^{|S| \cdot \gamma_{max}}$, and $\hat{e}^{|S| \cdot \gamma_{max}} \to 1$.

At this point, the matrix form of inequality constraints for linear programming problems:

$$
E \cdot \begin{bmatrix} \phi_1 \\ \phi_2 \\ ... \\ \phi_N \end{bmatrix} + \begin{bmatrix} e^{-c_{S_1}} & & & \\ & e^{-c_{S_2}} & & \\ & & ... & \\ & & & e^{-c_{S_{2^N}}} \end{bmatrix} \cdot \begin{bmatrix} \hat{e} \\ \hat{e} \\ ... \\ \hat{e} \end{bmatrix} \geq \begin{bmatrix} W_1 \\ W_2 \\ ... \\ W_{2^N} \end{bmatrix}
$$

Among them, $E$ is a sparse matrix with $2^N \times N$, $E_{i,j} = 1$ if and only if $a_i \in S_j$, the coefficient matrix of the parameter $\hat{e}$ is the diagonal matrix $\Gamma$. Then,

$$\Gamma \cdot \hat{e} = W - E \cdot \Phi \tag{18}$$

$$|\hat{e}| = \frac{\|W - E \cdot \Phi\|}{\|\Gamma\|} \tag{19}$$

$$\lim_{N \to \infty} \Delta |\hat{e}| = \frac{\|W - E \cdot \Phi\|}{\|\Gamma\|} - \|W - E \cdot \Phi\| \tag{20}$$

$$= \|W - E \cdot \Phi\| \cdot \frac{1 - \|\Gamma\|}{\|\Gamma\|} \tag{21}$$

$$\to 0 \tag{22}$$

We use the $L_2$-norm of matrix during the proof. Since the product of a bounded number and a number that converges to 0 still converges to 0, the Equation21 holds.

### C.2  MITIGATING THE EFFECTS OF IRRATIONALITY

The smaller the non-negative coefficient of $e$ in the inequality constraint is, the more extent the constraint is considered. In this section, we examined the impact of the implementation of the allegations above by the Number Shape Union Method. Taking the $DC$ with two agents for example, the solution of linear programming equation 9 always lies in the line of equality constraint (or the equation will have no solution).

Supposed that the least-core $e$ has already been computed, we then examined the impact on the allocation of $\Phi = (\phi_1, \phi_2)$. As is shown in three figures, the Red dotted line represent the restriction $\phi_i + c_i \cdot e \geq W_i$, and the blue solid line represents the equality constraint $\phi_1 + \phi_2 = \frac{B_0}{2}$.

**Case 1.** Concretely, the Figure 4 shows an ideal situation of the solution, since computing the equation16 brings the solution that $\phi_1 = \phi_2 = \frac{B_0}{2}$.

When the adjusted coefficients are apply to the inequality constraint, which can be seen as the red solid lines in figures, the optimal solution of equation16 may change. If not, the coefficients should satisfy $(1 - c_i) \cdot e \leq \frac{B}{2} - e$, equivalent to $c_i \geq \frac{B}{2e} - 2$. The assertion comes from the idea that the new inequality constraint becoming "closer to the optimal solution" but not overpass it. And the idea can be described as:

$$\phi_i' - \phi_i = (W_i - c_i \cdot e) - (W_i - e) \leq \frac{B}{2} - e, \quad i = 1, 2 \tag{23}$$

with $\phi_i'$ to be denoted as the new distribution solution.It means that the new inequality constraint would not overpass the original optimal solution, twisting the original allocation. Inversely, when it comes to the irrational expectation before compensating, the coefficient satisfies $c_i \leq \frac{B}{2e} - 2$ would change the irrational solution instead.

**Case 2.** In this case, the optimal solution without applying coefficients is located at the intersection of the constraint line formed by the expectation of one of the agents and the equality constraint line. Assume that the vertical axis is compensation for $phi_2$ and the horizontal axis is compensation for $phi_1$,, then the $phi_1$ is more than $phi_2$ before adjusting the coefficient of parameter $e$.

When this situation is caused by irrational expectations, that is, in fact, the contribution of $a_2$ is greater than that of $a_1$, but the final compensation is less because of the less expected compensation.

It is obviously that the inequality constraint boundary will be close to the parallel coordinate axis while the coefficient is greater than 1, otherwise away from the coordinate axis. Additionally, the new solution set merely influenced by the constraint line formed by the expectation of $a_1$ and the equality constraint line in this case. Therefore, $c_1$, the coefficient of parameter $e$ in $a_1$'s inequality constraint, is expected to have a larger coefficient while the equality constraint is fixed. This causes the new optimal solution to move closer to the axis perpendicular to $\phi_2$, increasing the compensation of it and decreasing the compensation of$\phi_1$.

**Case 3.** The case is the symmetric form of Case 2, and the same conclusion can be obtained by similar proofs.

Overall, the function of the coefficient should satisfy the following properties:

- Non-negativity. The Non-negative coefficient avoids changing the sign of parameter $e$, which in turn affects the allocation.
- Monotonicity. Monotonically decreasing coefficients with increasing contribution rate can alleviate distorted distribution

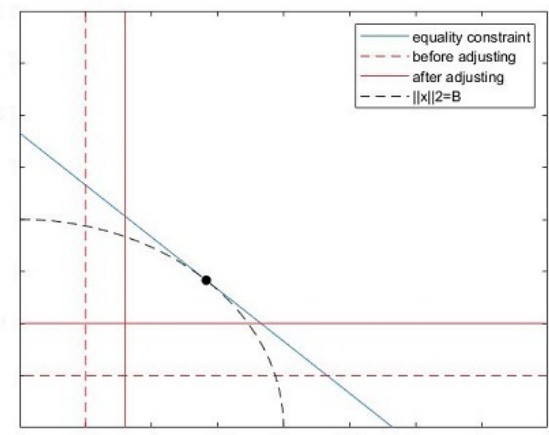

Figure 4: case1

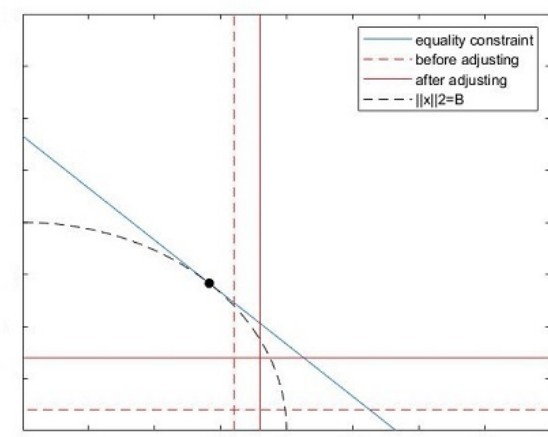

Figure 5: case2

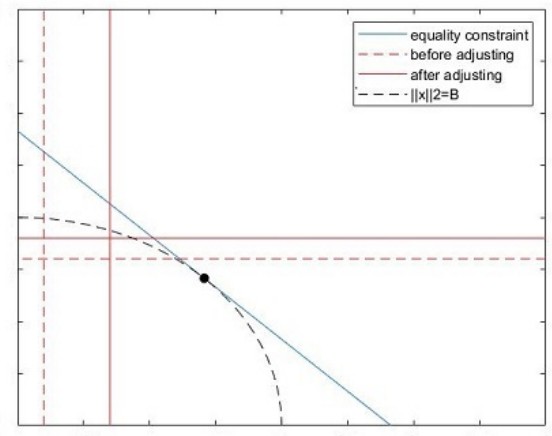

Figure 6: case3

