# OpenReview forum: "Data Pricing Mechanism Based on Property Rights Compensation Distribution"
_ICLR.cc/2023/Conference — Submitted to ICLR 2023_

### Official Review · Reviewer_V9Lr · 2022-10-23

**Confidence:** 3
**Clarity, Quality, Novelty And Reproducibility:** 1. How do you determine $W_i$ for eac…
**Correctness:** 3
**Technical Novelty And Significance:** 3
**Empirical Novelty And Significance:** 2
**Recommendation:** 8

**Strength And Weaknesses:**

Strength:
1. Provides data ownership from an economist’s perspective. It is an insightful contribution to formulate data ownership and pricing problem in the property right theory.
2. The time value introduction to contribution calculation is interesting.

Weakness:
1. The writing of the paper needs major improvement. Proofread to make the paper flow consistently. Also, providing brief explanations for concepts used as well as intuitive interpretations of the theoretical results could significantly improve the readability of the paper.
2. Some details of the method are not clearly described, partially due to the flow of the paper.
3. The experiments are lacking.

I have provided more detailed explanations of the weaknesses below.

**Summary Of The Paper:**

This paper discusses the data ownership problem from an economic perspective. The authors propose to use a compensation mechanism in the Modern Property Right Theory for solving data pricing problems in machine learning. Transferring the property right or ownership requires compensation, which is calculated by a modified least-core calculation. The formulation also incorporates fairness notions by introducing a parameter related to the contribution rate calculated through Shapley values. The paper also shows empirical experiments on data removal to demonstrate the usefulness of the proposed mechanism.

**Summary Of The Review:**

This paper is a nice addition to the machine learning data pricing and ownership literature from an economist’s perspective. It is new to me that an agent contributing the most could perform integration through compensation, and subsequently gain income from all integrated data property rights. This process could be efficient from a societal point of view. However, careless writing and the lack of clarity in the presentation affects the readability of the paper.

---

> ### Author Response · Authors · 2022-11-18
> **Reply to Reviewer V9Lr (Page 1)**
>
> $\textbf{Q1}$: How do you determine $W_i$ for each of the agents?
>
> $\textbf{A1}$: Sorry that the calculation of $W_i$ is not clear, we cross-referenced the process in the paper. In fact, the expected compensation $W_i$ is equivalent to the time value of the $a_i$. The irrational $W_i$ can be divided into two situations, which are reflected in the Section 5 "EMPIRICAL EXPERIMENT".
>
> $\textbf{Q2}$: It is unclear why stability is naturally preferred over fairness in the allocations discussed at the beginning of Section 3.3.
>
> $\textbf{A2}$: First of all, the stable allocation scheme (core) is a solution set, including the fair allocation scheme (Shapley) [1]. We believe that stability should be considered first, not because it is easier to consider stability first and then fairness, but for the following reasons:
> First, property rights integration is the formation of a big alliance. The successful integration is that the integrator holds the data ownership and the integrated agents are willing to lose the remaining control. If one of the agents is not willing to join the grand alliance, it will increase the negative externalities of the whole, so we hope that the grand alliance is stable.  Then, In this stable solution set, we further require the allocation to meet the expectations of both the integrator and the integrated party, which further reduces the scope of the solution set. Later, we found that the distribution that fully meets the needs of the integrated agents may violate fairness. In this case, we proposed to reduce the unfairness caused by irrational requirements as much as possible. Therefore, fairness needs to be considered, but it is not our primary consideration, but further improvement after satisfying stability.
>
> $\textbf{Q3}$: I am confused about the rounds of investments described in Section 3.3. Can an agent make multiple rounds of investments?  Please clarify this part and also the $γ$ used in Equation (10).
>
> $\textbf{A3}$: Multiple rounds of agent investment refer to multiple utilization of agent resources. Taking datasets as an example, training in different models can perform differently, and with the increase in training times, benefits will accumulate. However, data is time-sensitive, and its marginal contribution rate will be reduced when estimating the rate of return. Therefore, the return rate of the two adjacent investments of this agent cannot be $(1+\gamma_t^i)$. The compound interest rate is used to calculate that after t times investments. Compared with the initial principal, the return rate of the datasets is $(1+\frac{\gamma_t^i}{t})^t$. Time value refers to the total value of assets at time t (after t times investments). If we do not want to underestimate the value of assets, the number of investments tends to infinity, denoted as $lim\gamma_t^i=$$\gamma_0^i$. When the integrator estimated the compensation, we believe that he estimated the compensation based on the time value of the agent assets not less than any investment.
>
> $\textbf{Q4}$: I need clarification about the problem setting. The paper’s title restricts the method to data pricing problems where I suppose data samples are the resource that each agent invests in and collects. However, it does seem that the setting can be more general such as each agent can perform different functions in the process. Please clarify this point as it might mean wider applicability of the proposed method.
>
> $\textbf{A4}$: Your suggestion makes a lot of sense. We have reiterated the set of the problem in the INTRODUCTION section and our contribution in the CONCLUSION, and hope to extend this to a broader application context. We also answered similar questions in "Reply to ELSX" A3, hoping to answer your question.
>
> [1]	CHALKIADAKIS G, ELKIND E, WOOLDRIDGE M. Computational Aspects of Cooperative Game Theory [J]. Springer Berlin Heidelberg, 2011.

---

> > ### Author Response · Authors · 2022-11-18
> > **Reply to Reviewer V9Lr (Page 2)**
> >
> > $\textbf{Q5}$: There is confusion between the relaxation deficit parameter e in the relaxed core in (1) and the standard exponential number in the softmax function in (7). Also, is there any other stronger justification for using the softmax instead of the linear normalization? For example, how does it affect the experiments discussed in Section 5.2?
> >
> > $\textbf{A5}$: Good comment! We have exchanged the deficit parameter $e$ in the relaxed core for $d$. And keep the standard exponential number $e$ in the softmax function in (7) still. We prefer softmax to the linear normalization for the following two characters: First, a negative marginal contribution can be normalized to a positive one by the softmax function. This avoids a special case where linear normalization cannot be calculated when the Shapley value of two agents is 0.5, -0.5 respectively. Secondly, when the number of agents is small, the difference in the marginal contribution can be huge. The linear normalization will cause the value of b to be close to 0 or 1, while the softmax normalization can alleviate this phenomenon. When the number of agents is large, the two normalization methods are similar to each other. Taking into account the limited space, the result of experiments on synthetic datasets which illustrate the second feature of the softmax function will show in the camera-ready.
> >
> > $\textbf{Q6}$:  1)The figures do not strongly support the necessity of Core-cs, since the performance is not significantly different from Core in three of the four plots. 2) The effectiveness of the valuation is questionable on the breast cancer dataset (Figure 3), why does the accuracy increase after removing the first 10-20% of the highest valued data? 3) For the data pricing scheme to be practical, a much larger scale experiment with more complex datasets should be performed, e.g., on MNIST, CIFAR-10/100 or ImageNet.
> >
> > $\textbf{A6}$: Very good suggestions. Firstly, we are sorry that the performance of Core-cs is not that outstanding in the circumstem of only 40% of the data removed, so we increased the censored range of the image display to observe better results in a larger range. Secondly, we consider the anomalous increase in the cancer dataset to be  attributed to the characteristic of the dataset itself, and other experiments based on this dataset can also confirm this point [2]. In addition, the abnormality may be exacerbated by our use of nonlinear normalization to make agents with negetive marginal contribution obtain positive contribution rates. So we hope to follow your third suggestion and validate on more and larger datasets，which involves datasets (and the amount of data) that have not been tried in previous data pricing experiments. If possible, the specific experimental results will be reflected in camera-ready.
> >
> > $\textbf{Q7}$: There lacks an intuitive and clear description of the property right integration in the introduction and motivation section. I feel this is necessary to understand property right transfer as well the overall idea of the paper.
> >
> > $\textbf{A7}$: Great comment! We rewrite Introduction and point out that the goal of right confirmation needs to be realized by property rights integration. It restates the dual goal of data pricing that can achieve data confirmation through ownership compensation.
> >
> > $\textbf{Minor}$:  Thank the reviewer for careful reading. We apologize for typos, grammar mistakes, unclear notations and missing citations. They will be corrected such that the overall writing meet standards.
> >
> > [2]	YAN T, PROCACCIA A D. If You Like Shapley Then You'll Love the Core; proceedings of the AAAI, F, 2021 [C].

---

> > > ### Comment · Reviewer_V9Lr · 2022-11-26
> > > **Post-rebuttal**
> > >
> > > I would like to thank the authors for their responses. They are useful in clarifying my earlier doubts. Do consider including those explanations in future revisions of this paper, I believe it will make the writing even clearer. My questions are adequately addressed, but I hope the authors can include the additional experimental results in the camera-ready version as promised.
> > >
> > > Here are some suggestions/references to make the paper stand in the literature:
> > > - References for data-based pricing are missing. See model-agnostic pricing techniques [1,2,3].
> > > - As pointed out by other reviewers, you can include more relevant papers for the “Compensation distribution” in the ML setup. Refer to 1) the survey [4] for what they summarize and 2) other early works [5,6] for the papers that cited them.
> > > - For estimations of marginal contribution in specific models and setups, some papers potentially reduce negative externalities due to computation, refer to [7,8,9,10].
> > >
> > > Overall, I believe the paper carries significant novelty and presents a new perspective on data value from an economic perspective. I am very glad to see that the writing of the paper has been improved to meet the acceptance standard. Therefore, I would like to raise my score to accept the paper.
> > >
> > > [1] Validation-free and Replication Robust Volume-based Data Valuation, NeurIPS 2021.
> > >
> > > [2] Incentivising Collaboration in Machine Learning via Synthetic Data Rewards, AAAI 2022.
> > >
> > > [3] Fundamentals of Task-Agnostic Data Valuation, arXiv 2022.
> > >
> > > [4] Data Valuation in Machine Learning: “Ingredients”, Strategies, and Open Challenges, IJCAI 2022.
> > >
> > > [5] Understanding Black-box Predictions via Influence Functions, ICML 2017.
> > >
> > > [6] Collaborative Machine Learning with Incentive-Aware Model Rewards, ICML 2020.
> > >
> > > [7] Efficient Task-Specific Data Valuation for Nearest Neighbour Algorithms, VLDB 2019.
> > >
> > > [8] Improving Cooperative Game Theory-based Data Valuation via Data Utility Learning, ICLR Workshop 2022.
> > >
> > > [9] DAVINZ: Data Valuation using Deep Neural Networks at Initialization, ICML 2022.
> > >
> > > [10] Gradient Driven Rewards to Guarantee Fairness in Collaborative Machine Learning, NeurIPS 2021.
> > >
> > >
> > > Typo: Do not capitalize “Data” in your second contribution statement.

---

> > > > ### Author Response · Authors · 2022-12-10
> > > > **We appreciate your help in improving this paper**
> > > >
> > > > Dear Reviewer V9Lr,
> > > >
> > > > Thank you for your  valuable comments, we will happy to address your concerns accordingly and upload the revised version.
> > > >
> > > > In all, we would like to express our gratitude to the reviewer for being with us during this process and it is our pleasure to have this valuable experience! Your valuable feedback has a profound impact on the development of this paper and also on our understanding of this paper and broader issues. Your insights and comments have influenced the trajectory of our line of research as well as the development of our future work.
> > > >
> > > > Kind Regards,\
> > > > Authors of Paper857

---

### Official Review · Reviewer_ELSX · 2022-10-26

**Confidence:** 3
**Correctness:** 3
**Technical Novelty And Significance:** 4
**Empirical Novelty And Significance:** 3
**Recommendation:** 5

**Clarity, Quality, Novelty And Reproducibility:**

The clarity and quality of writing can be improved. Novelty is great. Reproducibility can be improved by providing some hyper parameters of the experiment.

**Strength And Weaknesses:**

Strength:

- The paper takes an inter-disciplinary approach and couples property right theory, cooperative game theory, machine learning to address the problem of data valuation

- The questions examined by the paper are crucial: how to address the ownership problem on a data chain? How to compensate each agent?

Weaknesses:
-Section 4.3: it’s not clear how to determine $B_0^i$ in Eqn (8)
- The efficiency aspect of the pricing mechanism in (9)-(10) needs to be further discussed. Specifically, to determine $\gamma$, one needs to solve Shapley value, which has exponential complexity in the number of agents. The optimization in (9)-(10) per se is also combinatorial in the number of agents.
- One of the most intriguing part of the proposed framework is the ability to deal with data chain (i.e., different agents take different roles in a data pipeline line: some are data pre-processor, some are model trainer). However, this part is not empirically validated in the experiment
- Section 5.1 aims at validating Theorem 1. However, this theorem was not the contribution of this paper but proposed in Yan et al. It’ll be great if the paper can focus on evaluating the differentiating factors of the paper.
- The writing of the paper can be significantly improved. This manuscript contains a lot of typos and grammar issues.
- If the paper’s final target venue is ML conference, I would suggest the authors to significantly revise Section 3 “Preliminaries and Background” to contextualize the property right theory and other economic concepts in the machine learning and data analytics setup.


**Summary Of The Paper:**

The paper proposes a data valuation framework based on property right theory.

**Summary Of The Review:**

The paper takes an interdisciplinary approach to address an important question in data market. However, the experiment part does not sufficiently validate the key contributions of the paper.

---

> ### Author Response · Authors · 2022-11-18
> **Reply to reviewer ELSX**
>
> We thank the reviewers for their careful consideration. We greatly appreciate the positive comments and address major concerns below.
>
> $\textbf{Q1}$: It’s not clear how to determine $B_0^i$ in Eqn (8).
>
> $\textbf{A1}$: Thank you for your suggestion, we will refine the explanation of this parameter and add quotation marks in the text. $B_0^i$ refers to the compensation that the consolidator willing to pay for the $i_{th}$ agent. It is the valuation given by the integrating party based on an estimate of the time value $C_t^i$ for $agent_i$ at time $t$. And the $C_t^i $ determined by Eqn(5) in text. Since it is reasonable for integrator that the compensation should be at least not less than the time value of the integrated party, i.e., satisfy $\lim_{t\to\infty}\frac{C_{t}^{i}}{B_{0}^{i}}=1$ ,the $ B_0^i$ is determined.
>
> $\textbf{Q2}$: The efficiency aspect of the pricing mechanism in (9)-(10) needs to be further discussed.
>
> $\textbf{A2}$: Thank you for your correction. We have modified the part of the discussion on computational efficiency in our paper. The computational complexity of the cs-core will mainly come from $B_0$, i.e., influenced by Shapley value. Even though we reduce the complexity by calculate the tmc-sv[1], it is still a huge work to estimate the SV. The future work might consider simpler estimation of the marginal contribution, such as using Cosine Gradient[2] as a uitility function to approximate SV. But the impact of this method needs to be further discussed and perhaps can be placed in future work to improve it.
>
> $\textbf{Q3}$: One of the most intriguing part of the proposed framework is the ability to deal with data chain. However, this part is not empirically validated in the experiment
>
> $\textbf{A3}$: Thank you very much for agreeing that pricing data chains is interesting. Actually，we hope to intuitively show the concept of agents and the costs in the data marketplace through the data chain. We extract the importance of clarifying ownership from the problem of valuing the process in the data life cycle. Then we provide an integration method in line with the overall income according to the modern property rights theory. And finally, distribute the credit based on ownership compensation. Such a pricing framework can evaluate the processes in the data chain, but it goes beyond that and, as reviewer V9Lr mentioned, can be applied to a wider context. The reason why we did not show relevant experiments is that the assessment of the data full cycle involves fewer stages (such as data pre-processor,model trainer, etc.). We hope to experiment on larger datasets to demonstrate the feasibility of our proposed method compared with the baseline. As for the work of pricing the data chain, we hope to put it in future work. Since the difficulty of it is not the amount of calculation, we hope to further study from other perspectives. To sum up,  we add the explanations in Section 1 "INTRODUCTION" and  Section 5 "EMPIRICAL EXPERIMENT" to prevent misunderstanding.
>
>
>
> $\textbf{Q4}$: Theorem 1.  was not the contribution of this paper but proposed in Yan et al. It’ll be great if the paper can focus on evaluating the differentiating factors of the paper.
>
> $\textbf{A4}$: Thank you very much for your suggestion, we will reduce the discussion of non-original principles and focus on our methods . To make our paraments more clear, we supplement and reference the  parameter calculation in the Section 4 "METHOD". In addition, we also answered the difference between our approach and Yan's approach in "reply to reviewerWvJ5" A1. The main point is that we use such a framework to solve the problem of data ownership determination, and this theory helps us to solve the allocation problem initially, on top of which we propose a modified version to alleviate the irrational expectation situation of the integrated parties.
>
> $\textbf{Q5}$: The writing of the paper can be significantly improved.
>
> $\textbf{A5}$:  We apologize for typos, grammar mistakes, unclear notations. They will be corrected such that the overall writing meet standards.
>
> $\textbf{Q6}$: If the paper’s final target venue is ML conference, I would suggest the authors to significantly revise Section 3 “Preliminaries and Background” to contextualize the property right theory and other economic concepts in the machine learning and data analytics setup.
>
> $\textbf{A6}$: Thanks for the helpful comments! We have updated Section 3 "Preliminaries and Background". Specifically, we supplement the generation and definition of data ownership with the property rights theory.
>
> [1]	GHORBANI A, ZOU J. Data Shapley: Equitable Valuation of Data for Machine Learning; proceedings of the 36th International Conference on Machine Learning (ICML), Long Beach, CA, F Jun 09-15, 2019 [C]. 2019.
>
> [2]	XU X, LYU L, MA X, et al. Gradient Driven Rewards to Guarantee Fairness in Collaborative Machine Learning [Z]//RANZATO M, BEYGELZIMER A, DAUPHIN Y, et al. 2021: 16104--17

---

> > ### Comment · Reviewer_ELSX · 2022-12-12
> > **Appreciate the responses**
> >
> > I'd like to thank the authors for their detailed responses. My primary concern about the lack of evaluation on data chains still remains. I am intrigued by the scope of the paper to value data chains and the proposed ideas. Data chain valuation is the key differentiating factor from the existing literature. The lack of evaluation will make the readers wonder whether the ideas proposed in the paper are indeed feasible. I agree with the reviewer that the data size might be a problem. To tackle this problem, I recommend that the authors consider a small number of agents, e.g., each agent holding a subset of training data, instead of a data point of a dataset. In that case,  the combinatorial computational overhead associated with data valuation would be manageable. Overall, I see a clear path for this paper to be published eventually. But I think the current paper is still in the preliminary stage. The evaluation of the proposed technique is central to demonstrating the value of the proposed ideas and making an impact.
> >
> >
> > ----
> > I would like to add that there is a paper on data valuation for data chains: https://arxiv.org/abs/2204.11131?context=cs.DB which might be helpful for setting up the evaluation.

---

### Official Review · Reviewer_WvJ5 · 2022-10-28

**Confidence:** 3
**Correctness:** 4
**Technical Novelty And Significance:** 3
**Empirical Novelty And Significance:** 4
**Recommendation:** 6

**Clarity, Quality, Novelty And Reproducibility:**

The paper is well-written, but it is a bit difficult to grasp the exact contribution of the work. The conceptual contribution is good, but the methodological contribution is limited based on the previous work. The contribution is original as far as I know.

**Strength And Weaknesses:**

Strength:
The paper tries to address an important problem through a new lens. The main contribution is the conceptual contribution of adopting the property rights theory in the problem of data valuation. Basically, they adopt the theory to give the agent with the highest contribution the power of integrating user data and allocating payments.

Weakness:
1. The main techniques come from previous literature. The method of deciding the contribution is basically the Data Shapley method from (Ghorbani & Zou, 2019), and the method of allocating compensations is just the least-core method from (Yan & Procaccia, 2021).
2. The motivation for using property rights theory to assign integration power is not clear. How is this method fundamentally different from having a trusted central planner? It seems to me that the method is basically choosing a central planner among the agents based on their contributions. Furthermore, it is not clear how to prevent the integrator from malicious behaviors of giving non-truthful payments.

**Summary Of The Paper:**

The paper proposes a data valuation mechanism based on modern property rights theory aiming to address the problem of data rights and data integration. The framework gives the power of integrating data and allocating compensations to the agent that contributes most significantly. Their framework uses a contribution evaluation method based on the Data Shapley to decide who has the power to integrate the data and give compensation. And they assign the compensation using the Least-core method. Finally, they demonstrate experimentally that the compensation mechanism is feasible in solving the data pricing problem.

**Summary Of The Review:**

The main contribution of the paper is the conceptual contribution of using the modern property right theory in the problem of data valuation. The methodological contribution is not super strong based on the previous work. The motivation and the practicability of their method are not clear.

---

> ### Author Response · Authors · 2022-11-18
> **Reply to Reviewer WvJ5**
>
> We thank the reviewer for affirmation of our concept innovation and the valuable comments.  We address all the points as follows.
>
> $\textbf{Q1}$: The main techniques come from previous literature. The method of deciding the contribution is basically the Data Shapley method from (Ghorbani & Zou, 2019), and the method of allocating compensations is just the least-core method from (Yan & Procaccia, 2021).Our experimental approach to test the pricing method is referenced from (Ghorbani & Zou, 2019), whose proposed experiments are so classical that they are the most common in verifying the validity of valuation approaches.
>
> $\textbf{A1}$: The basic framework of the allocation method is the least-core, a concept that has appeared in game theory for a long time and is neither our nor Yan's first. And Yan introduced it to the data pricing problem by assigning credit to agents that are no less than the utility in an arbitrary coalition state, while we use the framework to distribute compensation that satisfies the expectations of both the integrating party and the integrated party. Our application scenarios are different and solve different problems. Such different problem contexts make our allocations more susceptible to the influence of irrational compensation of the integrated party, thus, we make minor adjustments to the classical minimum core concept to reduce this influence. In summary, we refer to these two basic frameworks to respect the authority of the experimental approach on the one hand, and we make appropriate adjustments to the allocation method according to the research problem on the other hand.
>
> $\textbf{Q2}$: We will answer the second question by dividing it into two sub-questions:
>
> $\textbf{Q2.1}$: The motivation for using property rights theory to assign integration power is not clear. How is this method fundamentally different from having a trusted central planner?
>
> $\textbf{A2.1}$:First of all, our property consolidators come from the agents trading, rather than another trusted third-party platform. Secondly, the integrator integrates all agents involved in the transaction into a whole to achieve the purpose of maximizing the overall income（Profit minus transaction cost）. A trusted third party is similar to an intermediary, such as the Broker introduced in Dealer[1]. He can provide utility estimation for model buyers to choose the most suitable one for their needs, and protect the privacy of sellers through differential privacy. But this planner does not answer the question that who owns the property rights of data. At the same time, the agents in this framework have not been integrated, therefore, they only pursue the maximum personal interest rather than the overall optimal performance, so they have to bear more negative externalities.
>
> $\textbf{Q2.2}$: It is not clear how to prevent the integrator from malicious behaviors of giving non-truthful payments.
>
> $\textbf{A2.2}$:We thank the reviewer for the suggestion. It is supposed that the system for determining who integrates ownership among collaborators is open and transparent. It means that the collaborators know each other's benefits at the moment t (e.g., the financial reports of each company) so that theoretically there should be non-truthful payments. But it is worth considering to avoid such situations, which we will explore in future work.
>
> [1]	LIU J. Dealer: End-to-End Data Marketplace with Model-based Pricing [J]. ArXiv, 2020, abs/2003.13103.

---

> > ### Comment · Reviewer_WvJ5 · 2022-11-29
> > **Thank you for the response**
> >
> > I want to thank the authors for their response. It addresses my concern Q1. But I still find it difficult to grasp the main contribution probably due to the lack of background knowledge. It would be helpful if the authors could further clarify the following two questions.
> > 1. What will happen if I just randomly choose an agent to have the data ownership? What is the advantage of using property rights theory to assign integration power? Is it because we want to encourage people to contribute more? But it seems to me that people do not have incentives to withhold data even if we choose the integrator randomly, because they will get a higher compensation when they contribute more?
> > 2.  Regarding the response "At the same time, the agents in this framework have not been integrated, therefore, they only pursue the maximum personal interest rather than the overall optimal performance, so they have to bear more negative externalities." How does your framework prevent people from pursuing the maximum personal interest? Could you explain a bit more why they bear more negative externalities?

---

> > > ### Author Response · Authors · 2022-12-10
> > > **Response to  Reviewer WvJ5**
> > >
> > > We thank you for your comments and feedback. In addition to the later updates, we address your concerns here.
> > >
> > > $\textbf{Q1:}$ What will happen if I just randomly choose an agent to have the data ownership? What is the advantage of using property rights theory to assign integration power?
> > >
> > > $\textbf{A1:}$ This question relates to the motivation for our choice of modern property rights theory. The theory presents the idea that who takes ownership in the process of consolidation will have a critical impact on the outcome. There can be harmful effects associated with the wrong allocation of ownership. For instance, the agent with a lower marginal contribution taking ownership will lead to the agent with a higher marginal contribution losing the incentive, reducing their input, or even not input, and the final output decreases significantly. At this time, vertical integration is not as good as a market transaction.
> > >
> > > Overall, the advantage of consolidating property rights is to internalize the externalities and increase the overall output. Then the modern property rights theory proposes the agent with the larger marginal contribution to hold ownership, which makes the integration more effective. Finally, the distribution of compensation based on the contribution and the expectations of the integrated agents can make the integrated ones willing to transfer their ownership rights.
> > >
> > > $\textbf{Q2:}$ We will answer the second question by dividing it into two sub-questions:
> > >
> > > $\textbf{Q2.1:}$ How does your framework prevent people from pursuing the maximum personal interest?
> > >
> > > $\textbf{A2.1:}$ According to modern property rights theory, the ownership is referred to as residual control, i.e., the agent holding ownership to the subject matter has control over all exceptions except those agreed upon in advance in the contract. Therefore, the agent losing residual control can not make their decision after integration as well as pursue personal gain. And the integrating party makes the decisions and bears the overall gain or loss.
> > >
> > > $\textbf{Q2.2:}$ Could you explain a bit more why they bear more negative externalities?
> > >
> > > $\textbf{A2.2:}$ In terms of the model trainer, he will take on more computation when he is not integrated. Specifically, the agent needs to price the data every time he trains the model since even the same dataset performs differently in various models. By contrast, once the data ownership is integrated, the negative externalities of using the data can be reduced by reasonably estimating the ownership and making a one-time purchase.

---

> > > > ### Comment · Reviewer_WvJ5 · 2022-12-13
> > > > **Thank you for the response.**
> > > >
> > > > Thank you for the response.

---

### Decision · Program_Chairs · 2023-01-20

**Decision:**

Reject

**Justification For Why Not Higher Score:**

See the main concern (A).

**Justification For Why Not Lower Score:**

N/A.

**Metareview: Summary, Strengths And Weaknesses:**

The main contribution of this work lies in proposing a novel data valuation method based on property rights theory.

All the reviewers have agreed that the contributions of this work are novel and interesting.

On the flip side, there is a fairly large number of clarification and writing issues to be resolved. The authors have provided a detailed rebuttal to clarify most of them as well as revised the paper to rectify some writing issues. The authors are encouraged to do another proof-check to clear up the remaining issues with writing.

(A) The main concern shared by the reviewers during the discussion is the lack of empirical evaluation on data chain valuation which is the central idea/motivation of this work. In this regards, the authors need to set up appropriate experiments for empirically validating the performance and benefits of their proposed approach.

**Summary Of Ac-Reviewer Meeting:**

As mentioned in the meta-review, the reviewers in attendance have reiterated the novel and interesting contributions on this work. They have also understood this work better after the authors have revised the paper and provided clarifications in their rebuttal.

However, the reviewers also raise the aforementioned main concern (A) during the AC-reviewer meeting, which we all agree is the key issue that resulted in the final recommendation/decision.